Chitosan-DNA nanoparticles: synthesis and optimization for long-term storage and effective delivery

http://orcid.org/0000-0001-8722-5971 Raimbekova Aigul
http://orcid.org/0009-0004-2991-716X Kart Ulpan
Yerishova Akbayan
Elebessov Timur
http://orcid.org/0000-0002-7136-7921 Yegorov Sergey
Pham Tri Thanh
Hortelano Gonzalo gonzalo.hortelano@nu.edu.kz
Department of Biology, School of Sciences and Humanities, Nazarbayev University , Astana , Kazakhstan
Ravishankar Prashanth
Electronic publication date: 2025 Jan 24
Publication date: 2025
Volume: 13
Electronic Location ID: e18750
Received 2024 Sep 6; Accepted 2024 Dec 2
Copyright: © 2025 Raimbekova et al.
Copyright year: 2025
Copyright holder: Raimbekova et al.
License: This is an open access article distributed under the terms of the Creative Commons Attribution License, which permits unrestricted use, distribution, reproduction and adaptation in any medium and for any purpose provided that it is properly attributed. For attribution, the original author(s), title, publication source (PeerJ) and either DOI or URL of the article must be cited.
License URL: https://creativecommons.org/licenses/by/4.0/

Keywords: Chitosan nanoparticles, Lyophilization, Lyophilized chitosan nanoparticles, Transfection efficiency, H2B-mScarlet plasmid

Funding: Nazarbayev University, Kazakhstan (FDCRGP) 11022021FD2911 This research was funded by Nazarbayev University, Kazakhstan (FDCRGP grant 11022021FD2911) to Gonzalo Hortelano. The funders had no role in study design, data collection and analysis, decision to publish, or preparation of the manuscript.

==============================
Background

Chitosan nanoparticles (CsNPs) are an effective and inexpensive approach for DNA delivery into live cells. However, most CsNP synthesis protocols are not optimized to allow long-term storage of CsNPs without loss of function. Here, we describe a protocol for CsNP synthesis, lyophilization, and sonication, to store CsNPs and maintain transfection efficiency.

Methods

The size and zeta potential of CsNPs were analyzed by dynamic light scattering (DLS) and the morphology of CsNPs was assessed by transmission electron microscopy (TEM). HEK293 cells were transfected with CsNPs, and expression of H2B-CMV-mScarlet plasmid was assessed by flow cytometry. Confocal microscopy was used to visualize post-transfection gene expression. Time, volume, and effect of sonication were tested to optimize the lyophilization process.

Results

DLS and TEM analysis indicated amine groups on chitosan to phosphate groups on DNA (N:P) ratios yielded smaller CsNPs sizes. Transfection efficiency, measured by FACS and confocal microscopy, peaked at N:P ratios of 2:1 and 3:1 for both fresh and lyophilized CsNPs. Chitosan/DNA complexes remained stable in solution for at least 72 h at a ratio ≥2:1 as assessed by agarose gel electrophoresis. A lower surface charge with lower N:P ratios was indicated by zeta potential measurements. Lyophilized CsNPs lost 50% transfection efficiency compared to those freshly made. In contrast, sonication of lyophilized CsNPs restored their transfection efficiency to the level of fresh CsNPs. Sonicated CsNPs maintained spherical morphology, while unsonicated CsNPs showed aggregates. Cytotoxicity assays revealed high cell viability (>90%) after CsNPs transfection for a ratio of 2:1 or 3:1.

Conclusion

This optimized CsNPs synthesis protocol opens the possibility of long-term storage for CsNPs, which would provide broader applications of this technology.

Introduction

Nanoparticles generated through the process of salt-induced complex coacervation involving DNA and polycations have long been explored as DNA delivery vehicles (Li et al., 2016; Leong et al., 1998). Polymeric nanoparticles can be engineered to control characteristics such as size, surface charge, and morphology, tailoring them for specific tissues and applications (Zielińska et al., 2020; Wu et al., 2016). Chitosan nanoparticles (CsNPs) have emerged as a desirable option for DNA delivery vehicles because of their biodegradability, low toxicity, and biocompatibility. Chitosan, a polysaccharide from crustacean shells, is inexpensive and widely available. CsNPs leverage the beneficial properties of chitosan such as enhancing transcellular and paracellular transport across mucosal epithelium, which is particularly advantageous for gene delivery applications (Roy et al., 1999).

Despite their many advantages, CsNP technology faces challenges related to physical and chemical instability, particularly when stored in aqueous suspensions for extended periods (Dhadwar et al., 2010). Lyophilization (freeze-drying) is a promising method to address these issues, enhancing the stability of CsNPs by removing water from the frozen samples through sublimation and desorption under vacuum conditions. However, lyophilization can impose physical stresses such as ice crystal formation and solvent desorption, necessitating careful optimization of process parameters to mitigate these effects. According to previous studies, lyophilization can result in reduced transfection efficiency of nanoparticles (Veilleux et al., 2016), a significant concern for their application in DNA delivery (Leong et al., 1998). This is mainly due to water removal during the drying stage of the lyophilization process, which destabilizes the particles (Fonte, Reis & Sarmento, 2016) and form aggregation of NPs (Abdelwahed et al., 2006). Currently, cryoprotectants are been used to prevent the instability of lyophilized NPs. However, cryoprotectants need to be carefully optimized, because not all cryoprotectants are safe, effecatious and cost effective (Abdelwahed et al., 2006; De Giglio et al., 2023).

In this study, we describe a detailed protocol for CsNP synthesis, lyophilization, and sonication, demonstrating that this approach ensures higher nanoparticle stability, optimized size distribution, and increased their effectiveness in DNA delivery applications.

This article explores the critical parameters influencing the success of CsNPs freeze-drying and provides strategies for maintaining nanoparticle stability, thereby extending their shelf-life and enhancing their utility in DNA delivery applications.

Materials and Methods

Plasmid vectors

The CMV-H2B-mScarlet plasmid (Addgene, Watertown, MA, USA), harbourings a monomeric red fluorescent protein (Arendzen et al., 2023), was extracted by a MaxiPrep Kit (Qiagen, Hilden, Germany). The plasmid was aliquoted and stored frozen in EP buffer in a concentration at least of 1 μg/μL until needed.

Chitosan/DNA nanoparticle formulation

H2B-mScarlet CsNPs were formulated using the coacervation method (Roy et al., 1999). Chitosan CL113 (Mw ± 150 kDa) with a 90% degree of deacetylation (Novamatrix, Drammen, Norway) was used at a working concentration of 1 wt/vol % diluted with 25 mM acetic buffer, and H2B-mScarlet DNA was diluted in 50 mM sodium sulfate. Both solutions were heated to 55 °C for 20 min, mixed, and then vortexed for 30 s. After preparation nanoparticles were left to stabilize at room temperature for 30 min for stabilization. For the different DNA concentrations (0.005, 0.0025, and 0.00125 wt/vol %) various concentrations of chitosan were adjusted according to the different N:P ratios from 1:1 to 4:1.

Lyophilization of nanoparticles

Freshly prepared H2B-mScarlet CsNPs were placed into a plastic 50 ml tube and were freeze-dried under vacuum at −55 °C by Lyotrap Freeze Dryer (LTE Scientific, Greater Manchester, UK) and stored at 4 °C until used. Before being used in transfection experiments, lyophilized nanoparticles were reconstituted in nuclease-free water to the original concentration and sonicated for 10 min using a sonicator from Fisher Scientific, Model FB120 (Thermo Fisher Scientific, Waltham, MA, USA). The conditions used were 120 Watts and 20 kHz with an amplitude of 80 (Gokce et al., 2014). Subsequently, reconstituted nanoparticles were used as described for the freshly prepared nanoparticles.

TEM

The Jem-1200EX TEM (JOEL, Tokyo, Japan) set to operate at 80 kV was used to evaluate the morphology of CsNPs. A Formvar coated carbon 300 mesh coated Cu ++ grid (Ted Pella, CA, USA) was covered with a 10 μl solution of nanoparticles diluted 1:100 with nuclease-free water. The grid was further dried and viewed under TEM.

Particle size and zeta potential

The dynamic laser light scattering system (DLS) (Zetasizer, Malvern Instruments, Malvern, Worcestershire, UK) with a HeNe laser with a 635 nm wavelength was used to analyze the size and surface charge zeta potential (ZP) of nanoparticles. Scattered light was observed at a 90-degree angle. Nanoparticle samples were diluted in water to a 1:2,000 ratio and examined for 5 min, following the manufacturer’s instructions. Preparation and analysis of nanoparticles with various N:P ratios were conducted five times.

Nanoparticle stability

The integrity of the DNA released from nanoparticles was assessed by using gel electrophoresis. The 1% w/v agarose gel was stained with ethidium bromide (Sigma-Aldrich, Burlington, MA, USA). Loaded H2B-mScarlet CsNPs at 100 ng of DNA per well, and a single H2B-mScarlet DNA was used as control. Agarose gel was run for 45 min at 80 V and was visualized using a real-time UV transilluminator at 480 nm (Invitrogen, Carlsbad, CA, USA).

Transfection of HEK293 cells

HEK293 cells (ATCC, Manassas, VA, USA) were cultured in 24-well plates to 50% confluency with α-MEM, 10% fetal bovine serum, 1% Pen/Strep (Thermo Fisher Scientific, Waltham, MA, USA). HEK293 cells were transfected with CsNPs, DNA coupled to lipofectamine 2000 (LFN) (Invitrogen, Carlsbad, CA, USA) or unformulated DNA (-ve). For the transfection experiments, Opti-MEM (Gibco, Waltham, MA, USA) was used as the transfection medium, with adjusted to 6.5 using MES buffer (1 mM, pH 5.5) (Merck, Rahway, NJ, USA) (Ishii, Okahata & Sato, 2001).

Flow cytometry

FACS analysis was used to quantify H2B-mScarlet (ex/em wavelengths 569/594 nm) expression after CsNPs transfection. Briefly, HEK293 cells were washed thoroughly 24 and 48 hours (hrs) post-transfection three times with cold phosphate-buffered solution (PBS) (Sigma-Aldrich, Burlington, MA, USA) to remove unbound and surface-bound polyplexes. After the cells were trypsinized to detach cells (Therma Fisher, Waltham, MA, USA) for 3–5 minutes (mins) at 37 °C and then trypsin was neutralized with 10% Fetal Bovine Serum (FBS) Dulbecco’s Modified Eagle Medium (DMEM) media. Cells were collected and centrifuged and 4,000×g for 7 min and resuspended in PBS (Liang, Liu & Barman, 2019). Expression was analyzed using an Attune Nxt flow cytometer (Thermo Fisher Scientific, Waltham, MA, USA). The samples of 300 μL were collected to a designated 30,000 events.

Cytotoxicity of CsNPs

The viability of HEK293 cells before transfection was confirmed with an MTT (3-(4,5-Dimethylthiazol-2-yl)-2,5-Diphenyltetrazolium Bromide) assay (Thermo Fisher Scientific, Waltham, MA, USA). The protocol was adapted and modified (Riss et al., 2004). HEK293 cells were seeded in a 96-well plate (Thermo Fisher Scientific, Waltham, MA, USA) at a density of 2 × 104 cells/mL in 100 μL of DMEM medium. The cells were incubated overnight at 37 °C in a 5% CO2 atmosphere. After 24 h the cell medium was changed to H2B-mScarlet CsNPs in Opti-MEM and continued to be incubated for another 24 h. After adding 20 μL of 1× MTT solution to each well, the cells were incubated for an additional 4 h in an incubator (37 °C, 5% CO2 atmosphere). Following this, 100 μL of 10% sodium dodecyl sulfate (SDS) was added per each well and incubated overnight at room temperature. The absorbance at 570/630 nm was measured and expressed as relative values compared to the untreated negative control, which was considered 100% viable.

Confocal microscopy

HEK293 at a density of 1 × 105 cells per well were seeded in 8-well µ-Slide (Idibi, Bangkok, Thailand) and allowed to grow for 24 h. Four different ratios (1:1–4:1) of H2B-mScarlet CsNPs were then added to cells and incubated at 37 °C in a serum-free medium for 24 and 48 h. Images of cells expressing H2B-mScarlet were acquired using a Zeiss LSM980 confocal microscope (Zeiss, Oberkochen, Germany) and a 63× oil objective. Expression of H2B-mScarlet fluorescent emission was detected using an excitation of 569 nm with a multi-line Argon laser and detected through a 594 nm emission filter.

Statistical analysis

Statistical analyses were performed using GraphPad Prism 10. Normality tests were used to check whether the data satisfied the requirements of a normal distribution assumption for a parametric test. Depending on the normality of the data, the differences between experimental groups were compared using both parametric and non-parametric tests. FlowJo software was used to analyze data output.

Results

Size and morphology of CsNPs

The cellular uptake and internalization of nanoparticles are determined by their size, shape, surface charge, and surface functionalization, as well as the interactions among these factors. Therefore, achieving optimal cellular uptake efficiency requires attention to these factors at every stage of the process. The average particle size, PDI (polydispersity index), and PDI width of these CsNPs 0.0025 wt/vol % DNA (5 µg per 200 µl of nanoparticles) are listed in Table 1. DLS analysis showed that the size of CsNPs is dependent on the N:P ratio, the average particle size became larger with the increasing of chitosan concentration. The mean diameter of freshly prepared particles formulated with N:P ratios 1:1 and 2:1 was <300 nm (250 ± 12.9 and 290.6 ± 16.6 nm, respectively), while the higher N:P ratios of fresh particles (3:1 and 4:1) were sized at 349.5 ± 22.4 and 363.3 ± 32.8 nm, respectively.

Table 1 The average size, PDI and PDI width of CsNPs at different ratios (1:1, 2:1, 3:1 and 4:1).

		Diameters		
	CsNPs ratio	Z-average size (nm) ± SD	Polydispersity index (PDI) (mV) ± SD	PDI width (mV) ± SD	
Fresh	1:1	245.9 ± 12.9	0.169 ± 0.099	123.86 ± 4.07	
2:1	290.6 ± 16.6	0.261 ± 0.062	159.10 ± 18.86	
3:1	349.5 ± 22.4	0.263 ± 0.046	186.18 ± 18.89	
4:1	396.3 ± 32.8	0.303 ± 0.055	205.66 ± 41.44	
Lyophilized sonicated	1:1	246.1 ± 28.8	0.272 ± 0.099	148.30 ± 10.25	
2:1	276.2 ± 18.0	0.289 ± 0.028	154.28 ± 14.71	
3:1	354.6 ± 20.1	0.297 ± 0.043	215.98 ± 11.44	
4:1	383.5 ± 26.2	0.321 ± 0.044	247.30 ± 12.26	
Lyophilized unsonicated	1:1	439.4 ± 17.2	0.354 ± 0.017	254.08 ± 10.84	
2:1	470.5 ± 11.0	0.362 ± 0.015	354.86 ± 11.87	
3:1	547.3 ± 13.5	0.452 ± 0.007	549.50 ± 11.86	
4:1	915.1 ± 23.3	0.639 ± 0.013	640.00 ± 47.74	
Note:

Polydispersity index, mean size and PDI width of nanoparticles determined by DLS (n = 5) ± SD.

Lyophilization of nanoparticles is an attractive strategy for storage and medical applications. However, our group and others (Dhadwar et al., 2010) reported a drop in transfection efficiency of lyophilized (lyo) nanoparticles of approximately 50%. In this study, we studied whether the sonication of lyo nanoparticles could improve transfection efficiency. Fresh CsNPs had a comparable size range as lyophilized sonicated (lyo S) nanoparticles at the N:P ratio 1:1, 2:1, 3:1 and 4:1 mean sizes were 246.1 ± 28.8, 276.2 ± 18.0, 354.6 ± 20.1 and 383.5 ± 26.2. In comparison, lyophilized unsonicated (lyo Us) nanoparticles had a size range 2–3 times higher, which increased beyond 915.1 ± 23.3 nm as the N:P ratio increased to 4:1. Subsequently, the PDI of fresh and lyophilized sonicated particles at different ratio were 0.169 ± 0.099, 0.261 ± 0.062, 0.263 ± 0.046, 0.303 ± 0.055 mV and 0.272 ± 0.099, 0.289 ± 0.028, 0.297 ± 0.043, 0.321± 0.044 mV, respectively (Table 1). As the same size, the PDI of lyophilized unsonicated nanoparticles was increased beyond 0.639 ± 0.013 (Table 1). Those results show the effect of ultrasonication on the mean size and polydispersity of nanoparticles.

Next, TEM was utilized to assess the morphology of nanoparticle formulations made with chitosan CL113 at 1:1, 2:1, 3:1, and 4:1 N:P ratios, and H2B-mScarlet plasmid generated with different ionic strengths. We assessed the shape and size of fresh, lyophilized sonicated and lyophilized unsonicated nanoparticles (Fig. 1). Both fresh and lyophilized sonicated particles appeared globular in shape, which were constant across the ratios 1:1 to 4:1. Nanoparticles which were lyophilized but unsonicated had physical aggregations in all N:P ratios (Fig. 1).

Figure 1 The morphology of CsNPs.

Images taken using transmission electron microscopy of DNA (H2B-mScarlet) and chitosan (CL113) mixed at a N:P ratio.

Stability of CsNPs and DNA integrity

The stability of CsNPs was analyzed by measuring ZP and mean size for 24, 48, and 72 h. ZP quantifies effective electric charge on the surface of the nanoparticle. The magnitude of the ZP correlates with the stability of particles in suspension (Othman et al., 2012). Nanoparticles with higher ZP indicate that particles have a strong electrostatic repulsion between them, which is crucial in enhancing their stability by reducing the aggregation (Pardeshi et al., 2023). DLS analysis showed that the ZP of CsNPs was less positive with an N:P ratio of 1:1 (11.41 mV) and 2:1 (21.336) due to the low concentration of cationic Cs. Indeed, the addition of more chitosan resulted in more cationic nanoparticles. The electric repulsion induced by high ZP (>30 mV) can contribute to avoiding the aggregation of nanoparticles. Moreover, ZP is a key parameter for the stability of NPs in aqueous media. The ZP value was observed to be higher than 30 mV for the N:P ratios 3:1 (32.19 mV) and 4:1 (33.36 mV) for both fresh and lyophilized sonicated nanoparticles (Table S1) at 72 h.

Secondly, stability also was assessed by the DNA binding capacity of CsNPs, and 1% gel electrophoresis was performed at 24, 48, and 72 h as well. The binding capacity and stability of the CsNPs are reflected by the presence of unbound DNA in the gels. At all three time points, unbound DNA was observed in the fresh nanoparticles at the N:P ratio 1:1, indicating the ability of chitosan to form a stable complex with DNA in the N:P ratio 2:1 and (Fig. 2). However, no unbound DNA was detected in all ratios of both lyophilized sonicated and unsonicated nanoparticles. Our results confirmed the stability of CsNPs across all tested ratios and formations (Fig. 2).

Figure 2 DNA stability in chitosan.

DNA complexation with chitosan is determined using a gel shift experiment. H2B-mScarlet DNA, lane 1. The fresh, lyophilized sonicated and lyophilized not sonicated nanoparticles with different ratios represented by the lanes beneath each white line are 1:1, 2:1, 3:1 and 4:1 at 24, 48 and 72 h.

Transfection efficiency

The effect of the N:P ratio and DNA concentration on transfection efficiency was assessed by quantifying the level of H2B-mScarlet expression. HEK293 cells were transfected with nanoparticles formulated with H2B-mScarlet DNA using an equal volume of nanoparticles per 4 × 105 cells per well, and H2B-mScarlet expression was quantified 48 h post-transfection via flow cytometry. Nanoparticles at 2:1 and 3:1 N:P ratios showed the highest gene expression in all tested concentrations (0.00125, 0.0025, and 0.005 wt/vol %). The difference between N:P ratios of 2:1 and 3:1 was not statistically significant (p > 0.01). Compared to the lowest (0.00125 wt/vol %) and highest concentrations (0.005 wt/vol %), cells transfected with the 0.0025 wt/vol % at all four N:P ratios also demonstrated the highest mean H2B-mScarlet expression levels. N:P ratios of 2:1 and 3:1 showed the highest levels of gene expression, achieving 27.10% and 25.98%, respectively.

In contrast, cells transfected with nanoparticles at a 1:1 ratio exhibited the lowest gene expression 16.81% (Fig. 3). These results indicated a positive dose-response relationship, revealing a direct correlation between nanoparticle dose and transgene expression for the same formulation. Confocal microscopy confirmed visually that the CsNPs at ratios 2:1 and 3:1 have the highest level of transfection efficiency (Fig. S1).

Figure 3 Effects of N:P ratios on the H2B-mScarlet expression in transfected HEK293 cells.

Chitosan was complexed with H2B-mScarlet DNA at different ratios from 1:1 to 4:1. Transfection levels were assessed by flow cytometry and represented in percentage of cells expressing H2B-mScarlet fluorescence. LFN stands for Lipofectamine, -ve stands for negative control. Data are presented as mean ± SD (n = 3). The level of statistical significance was assessed by one-way ANOVA, where **: 0.001 < p < 0.01, ****: p < 0.0001 and ns indicated not significant.

Further optimization of CsNP synthesis

Following our previous findings, we focused our optimization efforts on lyophilized nanoparticles at this specific N:P ratio of 2:1, as a ratio with high efficiency and low NPs size. Figure 4A presents the transfection efficiency of nanoparticles lyophilized for different time intervals (12, 24 and 48 h), with the 24-h time point yielding the highest efficiency at 28.46% (p < 0.0001). Figure 4B illustrates the effects of varying volumes during lyophilization (1,500, 3,000, 4,500 µL), where the 3,000 µL volume resulted in the highest H2B-mScarlet expression at 28.10% (p < 0.0001). Figure 4C depicts the impact of sonication on lyophilized nanoparticles before transfection, showing that lyophilized sonicated nanoparticles achieved nearly double the H2B-mScarlet expression (27.49%) compared to lyophilized unsonicated nanoparticles (12.62%).

Figure 4 Optimization of the lyophilized nanoparticles at a N:P 2:1 ratio.

(A) Effect of the duration of lyophilization process. (B) Effect of volume of nanoparticles (batch) sample being lyophilized for 24 h. (C) Effect of sonication. LFN stands for Lipofectamine, -ve stands for negative control. Data are presented as mean ± SD (n = 3). The level of statistical significance was assessed by one-way ANOVA, where ****: p < 0.0001 and ns indicated not significant.

Following the optimization, lyophilized nanoparticles were compared with fresh and lyophilized unsonicated nanoparticles at various N:P ratios of 1:1, 2:1, 3:1, and 4:1. Lyophilized sonicated nanoparticles showed the highest expression levels at ratios 2:1 and 3:1, 26.22% and 24.32%, respectively. Visual assessment on Confocal microscopy demonstrated similar results (Fig. S2). The ratios 2:1 and 3:1 of lyophilized sonicated nanoparticles demonstrated significantly higher transfection efficiency than unsonicated lyophilized nanoparticles (p < 0.0001) (Fig. 5). However, there was no significant difference between fresh and lyophilized sonicated nanoparticles at ratios 2:1 and 3:1. Specifically, at N:P ratio 2:1 the expression level of lyophilized unsonicated nanoparticles (12.05%) is approximately half that of the fresh (25.56%) and sonicated lyophilized particles (26.22%).

Figure 5 Transfection efficiency of fresh, sonicated lyophilized and unsonicated lyophilized nanoparticles in HEK293 cells.

Chitosan was complexed with H2B-mScarlet DNA at different ratios from 1:1 to 4:1. Transfection levels were assessed by flow cytometry and represented in percentage of cells expressing H2B-mScarlet fluorescence. LFN stands for Lipofectamine, -ve stands for negative control. Data are presented as mean ± SD (n = 3). The level of statistical significance was assessed by one-way ANOVA, where *: 0.01 < p < 0.05, ***: 0.0001 < p < 0.001, ****: p < 0.0001 and ns indicated not significant.

Cytotoxicity

The toxicity of CsNPs was examined by MTT assay that included the transfection of HEK293 cells with different ratios of fresh and lyophilized sonicated nanoparticles. The results reveal that all formulations in both states (fresh and lyo S) showed high cell viability at 72 h (Fig. 6). In ratios with the highest transfection efficiency, 2:1 and 3:1, the viability of fresh and lyophilized sonicated nanoparticles was 93.46% and 95.87% for the 2:1 ratio, 90.02% and 85.97% for the 3:1 ratio, respectively (Fig. 6A). We also tested different nanoparticles made with different concentrations of DNA at 2:1 N:P ratio (Fig. 6B). No significant toxicity of nanoparticle complexes for HEK293 cells was observed at concentrations of DNA under 0.005 wt/vol %, with viability of 83.65 % in fresh and 85.52% in lyophilized sonicated nanoparticles. For higher concentrations of DNA, viability was above 58.68% in both fresh and lyophilizedsonicated nanoparticles (Fig. 6B). The effect of N:P ratio and DNA concentrations on cell viability was also tested at 24 and 48 h (Figs. S3A and S3B). Similarly, at 72 h cell viability depends on the ratio and concentration of nanoparticles. In other words, the ratios from 1:1–4:1 at concentration 0.0025 wt/vol % were safe and non-toxic for HEK293 cells at 24, 48, and 72 h (Figs. S3A and S3B).

Figure 6 Cytotoxicity of fresh and lyophilized sonicated CsNPs.

HEK293 cells were treated by formulated freshly prepared and lyophilized sonicated nanoparticles for 72 h. (A) Different N:P ratios affect toxicity for concentration 0.0025%. (B) Different concentrations of NPs (from 0.00125% to 0.02%) at a ratio 2:1 on cell viability after 72 h. LFN stands for Lipofectamine, -ve stands for negative control. Data normalization was performed by considering the viability of untreated cells as 100%. Data represent mean ± SD (n = 5). The level of statistical significance was assessed by one-way ANOVA, where ****: p < 0.0001 and ns indicated not significant.

Discussion

Here we present an optimized protocol for CsNP synthesis, which ensures CsNP stability and high efficiency in DNA delivery applications. Our results demonstrate that CsNPs can be effectively synthesized and stored without loss of functionality by implementing lyophilization and sonication as part of the synthesis protocol.

A key factor we assessed was the impact that lyophilization time and nanoparticle batch volume had on transfection efficiency. Most published studies reported lyophilization duration of CsNPs at the range of 12–48 h (Adwan, Obeidi & Al-Akayleh, 2024; Angarita et al., 2024; Duarte Junior et al., 2017; Gutiérrez-Ruíz et al., 2024). According to our results, a lyophilization time of 24 h was the best, resulting in higher transgene expression than lyophilization conducted for 12 or 48 h. We hypothesized that 24 h would be optimal because it provides a balance between adequate dehydration and prevention of structural degradation, which can occur during prolonged drying. These findings align with previous studies by Ebrahimnejad et al. (2023) and Al-nemrawi, Alsharif & Dave (2018), who reported that lyophilization time is a critical factor in optimizing transfection efficiency.

Batch volume, closely related to lyophilization time, is another essential factor influencing the lyophilization process, as reposted by Jameel et al. (2021). Our findings indicate that, for our CsNPs, 3,000 µL is the optimal volume for 24 h lyophilization, demonstrating greater effectiveness than the 1,500 and 4,500 µL batches. We propose that this volume achieves optimal conditions during lyophilization, potentially allowing for more uniform dehydration. Since batch size is important for the industrial scale-up of nanoparticle manufacturing, further research could identify strategies to increase the volume of each production batch of CsNPs.

Introducing a sonication step after lyophilization was found to be the key factor for enhancing DNA expression levels, resulting in a two-fold increase in transgene expression, as confirmed by FACS analysis. These findings align with our hypothesis that smaller CsNP sizes are more likely to improve cellular internalization and transfection efficiency in DNA delivery systems. Lyophilized sonicated nanoparticles had a relatively smaller size and spherical morphology compared to lyophilized unsonicated particles, which exhibited significant aggregation and increased polydispersity, consistent with previously reported results (Gokce et al., 2014; Kaleemuddin & Srinivas, 2012; Braim et al., 2022), although those prior studies did not analyze cell transfection.

The N:P ratio, representing the amine groups on chitosan to phosphate groups on DNA is an important parameter influencing CsNP stability and performance. We found that lower N:P ratios yielded smaller particle sizes, which are generally favorable for cellular uptake (Aiping et al., 2006). However, the transfection efficiency peaked at N:P ratios of 2:1 and 3:1 for both fresh and lyophilized sonicated CsNPs, indicating that these ratios offer the best balance between DNA condensation and nanoparticle stability. This observation aligns with previous research, such as that by Erbacher et al. (1998), suggesting that moderate N:P ratios are optimal for gene delivery (Veilleux et al., 2016).

Furthermore, CsNPs with higher ZP values, observed at N:P ratios of 3:1 and 4:1, demonstrated enhanced stability, likely due to stronger repulsive forces that prevent aggregation (Zhu et al., 2005). Nanoparticles that are positively charged are more likely to interact with negatively charged cell membranes, facilitating DNA release into the cytoplasm (Simunkova et al., 2009; Erbacher et al., 1998). Gel electrophoresis confirmed the stability of CsNPs, with free DNA only detected in the 1:1 ratio of fresh nanoparticles, corroborating previous findings from our laboratory (Zielińska et al., 2020).

Importantly, our study demonstrated high cell viability (up to 95%) across all formulations and N:P ratios at 72 h, with the 2:1 (95% viability) and 3:1 (90% viability) ratios showing particularly favorable outcomes. Even at higher DNA concentrations and after 72 h of exposure, we observed minimal cytotoxic effects of CsNP compared to LFN, a commercial transfection reagent. These results are consistent with the high biocompatibility of CsNPs (Yuan et al., 2009).

The choice of N:P ratios may depend on the specific application, as it influences cell viability and cytotoxicity. For example, an N:P ratio of 3:1 has been found to show slightly higher cytotoxicity than a 2:1 ratio, likely due to the increased nitrogen presence impacting cellular health at higher concentrations. Additionally, ZP measurements tend to rise with higher N:P ratios, which can influence particle stability and cellular uptake in solutions.

Slight differences in cytotoxicity and zeta potential were observed at an N:P ratio of 3:1 compared to 2:1. Cytotoxicity was slightly lower, while the zeta potential was slightly higher for nanoparticles at this N:P ratio.

Our study has limitations. Firstly, the long-term stability of CsNPs under varying storage conditions was not fully explored. Secondly, the transfection efficiency was only tested in HEK293 cells, and future studies need to be done on different cell lines and in vivo models. Despite these limitations, our study presents a reliable and reproducible protocol for preparing, lyophilizing, and recovering CsNPs, which ensures their stability and high transfection efficiency. The novelty of our study lies in its comprehensive assessment of transfection efficiency across all optimization parameters in vitro, evaluated using flow cytometry. By systematically optimizing and testing N:P ratios, lyophilization conditions, and post-lyophilization sonication, this study provides a detailed and quantitative understanding of how each protocol step impacts DNA delivery efficiency, setting this work apart from previous studies. We hope these results will facilitate the development of CsNP-driven applications for gene delivery, particularly in scenarios requiring extended storage and rapid deployment.

Conclusion

This study systematically elucidates the interplay between N:P ratio, sonication, and lyophilization on their effect on the performance of CsNPs. The ratios 2:1 and 3:1 consistently delivered the best outcomes regarding particle size, stability, low cytotoxicity, and high transfection efficiency. These findings provide a robust framework for optimizing CsNPs, paving the way for their effective use in gene therapy and other biomedical applications. Future research should explore the in vivo performance of these optimized nanoparticles to further validate their therapeutic potential. This study suggests that a CsNPs preparation protocol including lyophilization plus sonication may allow the long-term storage of CsNPs while maintaining transfection efficiency, thus making medical applications of CsNPs feasible.

Supplemental Information

Supplemental Information 1 Raw data for all tables and figures.

Supplemental Information 2 The original agarose gel images.

Supplemental Information 3 Size distribution of fresh and lyophilized sonicated CsNPs, analyzed with dynamic light scattering.

Different ratios (1:1-4:1) of CsNPs were treated with DMEM for 24, 48 and 72 hours as mean (n=5) ±SD. During the time and different ratios, the mean size of the particles was not changed much, indicating CsNPs were stable during the time.

Supplemental Information 4 Confocal laser scanning microscopy of HEK293 cells, transfected for 48 hours with different ratios (1:1-4:1) of H2B-mScarlet CsNPs.

LFN stands for Lipofectamine, -ve stands for negative control.

Supplemental Information 5 Confocal laser scanning microscopy of HEK293 cells transfected for 48 hours with different ratios of lyophilized nanoparticles (1:1-4:1) of H2B-mScarlet CsNPs.

LFN stands for Lipofectamine, -ve stands for negative control.

Supplemental Information 6 Cytotoxicity of CsNPs.

HEK293 cells were treated with CsNPs for 24 and 48 hours. Normalization of the data performed based on considering the viability of untreated cells as 100%. LFN stands for Lipofectamine, -ve stands for negative control. Data are mean ± SD (n=5). The level of statistical significance was assessed by One-Way ANOVA, where *: 0.01<p<0.05, **: 0.001<p<0.01, ***: 0.0001<p<0.001, ****: p<0.0001 and ns indicated not significant.

Supplemental Information 7 DOI for flow cytometry data.

We thank Dr. Anna Andreeva for sharing plasmids, as well as Mrs. Aigul Kussanova and Mrs. Nurgul Daniyeva for their excellent technical support.

Additional Information and Declarations

Competing Interests

The authors declare that they have no competing interests.

Author Contributions

Aigul Raimbekova conceived and designed the experiments, performed the experiments, analyzed the data, prepared figures and/or tables, authored or reviewed drafts of the article, and approved the final draft.

Ulpan Kart conceived and designed the experiments, performed the experiments, analyzed the data, prepared figures and/or tables, authored or reviewed drafts of the article, and approved the final draft.

Akbayan Yerishova performed the experiments, authored or reviewed drafts of the article, and approved the final draft.

Timur Elebessov performed the experiments, prepared figures and/or tables, and approved the final draft.

Sergey Yegorov analyzed the data, authored or reviewed drafts of the article, and approved the final draft.

Tri Thanh Pham conceived and designed the experiments, analyzed the data, authored or reviewed drafts of the article, and approved the final draft.

Gonzalo Hortelano conceived and designed the experiments, authored or reviewed drafts of the article, and approved the final draft.

Data Availability

The following information was supplied regarding data availability:

The raw data is available in the Supplemental Files.

The flow cytometry data is available in Zenodo:

Raimbekova, A. (2024). Figure 3 Effect of N:P ratios on the H2B-mScarlet expression in transfected HEK293 cells. [Data set]. Zenodo. https://doi.org/10.5281/zenodo.13709923

Raimbekova, A. (2024). Figure 4. Optimization of the lyophilized nanoparticles at a N:P 2:1 ratio [Data set]. Zenodo. https://doi.org/10.5281/zenodo.13710021

Raimbekova, A. (2024). Figure 5. Transfection eûciency of fresh, sonicated lyophilized and unsonicated lyophilized nanoparticles in HEK293 cells. [Data set]. Zenodo. https://doi.org/10.5281/zenodo.13710101.

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
