# Peer review of "Chitosan-DNA nanoparticles: synthesis and optimization for long-term storage and effective delivery"

_PeerJ, doi:10.7717/peerj.18750_

## Round 0.1 · original submission · Major Revisions

The authors are requested to address the reviewers' comments to improve the overall quality and clarity of the manuscript. Additionally, I recommend that the authors identify the major studies related to this study as highlighted by one of the reviewers, and the compare their data with the other studies. Provide references highlighting why the reported results are the same or better to previously reported work.

·

Basic reporting

General Comments
The manuscript addresses an important topic, providing valuable data on DNA-based delivery vehicles. However, to enhance accessibility and impact, improvements in text clarity, introduction structure, and discussion are recommended. While many of the results are well-presented, certain sections would benefit from a clearer rationale or hypothesis to strengthen the overall findings and make the manuscript more comprehensive.
Major Comments
• Abbreviation Clarifications:
o Comment: Some abbreviations are not fully explained when first mentioned, which could confuse readers who are unfamiliar with the terms.
o Suggested Action: Ensure that abbreviations are spelled out when introduced. For example, on line 35, the N:P ratio can be presented as "Nitrogen: Phosphorus ratio or chitosan amine: DNA phosphate (N:P) ratio." In Figure 4, "LFN" appears without explanation; Although it's later described, lipofectamine should be mentioned earlier for clarity.
• Hypothesis and Rationale in Conclusions:
o Comment: When discussing your results, it's important to state the rationale or hypothesis behind the results clearly. Comparative analysis with prior studies needs to be more explicitly framed.
o Suggested Action: For instance, the discussion in lines 275-285 would benefit from including a concise hypothesis explaining why your findings either align with or differ from prior research. Providing a clear rationale for the experimental outcomes will further strengthen and clarify this section.
• Introduction Expansion:
o Comment: The introduction requires more depth, particularly when discussing the preservation challenges of DNA-based delivery vehicles, such as lyophilization.
o Suggested Action: Consider expanding this section by including references to more studies that highlight issues with lyophilization, such as reduced transfection efficiency or DNA degradation. Additionally, examples of other methods used to optimize preservation protocols, along with their limitations, would provide valuable context. For example, studies like [Mao, H.Q., Roy, K., Troung-Le, V.L., Janes, K.A., Lin, K.Y., Wang, Y., August, J.T. and Leong, K.W., 2001. Chitosan-DNA nanoparticles as gene carriers: synthesis, characterization and transfection efficiency. Journal of controlled release, 70(3), pp.399-421.)] highlight alternative methods to optimize protocols. Presenting a brief hypothesis about why sonication offers an improved or comparable alternative, supported by underlying principles, would strengthen your argument.
o Additional References: Consider adding more references to support your points in lines 48-51.
Minor Comments
• Grammar and Readability:
o Comment: Certain sections could be grammatically improved for clarity.
o Suggested Action: Review lines 30-31, 208-210, and 300 for better readability and smoother comprehension.
Conclusion
Overall, this manuscript presents valuable data, but it would benefit from adding more depth to the introduction, providing clearer rationales for certain results, and enhancing clarity in a few sections. Focusing on these key areas will help improve the manuscript’s quality and readability.

Experimental design

No comments

Validity of the findings

No comment

·

Basic reporting

In the present manuscript the authors have reported synthesis of chitosan nanoparticles using coacervation method followed by lyophilization and sonication and transfections. The results are encouraging and data presented well. However, the manuscript lacks novelty.

Experimental design

The experiments are well conducted.

Validity of the findings

In the present manuscript the authors have reported synthesis of chitosan nanoparticles using coacervation method followed by lyophilization and sonication and transfections. The results are encouraging and data presented well. However, the manuscript lacks novelty.

Additional comments

In the present manuscript the authors have reported synthesis of chitosan nanoparticles using coacervation method followed by lyophilization and sonication and transfections. The results are encouraging and data presented well. However, the manuscript lacks novelty. The problem or the knowledge gap addressed in the study does not provide any novel information or add anything new to the existing information on the topic. Various studies have been conducted, which have already shown the use of lyophilization and sonication as the method to increase the stability of chitosan or different polymeric nanoparticles. Since, the infomtion is not new/novel and the study already available, I recommend to reject the manuscript from publication.
There are some of the previously published articles that deal with the same knowledge gap that is proposed in the study and have been published quite early as well.
1. Yavuz Gokce, Burcu Cengiz, Nuray Yildiz, Ayla Calimli, Zeki Aktas, Ultrasonication of chitosan nanoparticle suspension: Influence on particle size, Colloids and Surfaces A: Physicochemical and Engineering Aspects, Volume 462, 2014, Pages 75-81, ISSN 0927-7757, https://doi.org/10.1016/j.colsurfa.2014.08.028 mentions the effect of lyophilization and sonication on particle size and stability of chitosan nanoparticles.
2. Almalik, A., Alradwan, I., Kalam, M. A., & Alshamsan, A. (2017). Effect of cryoprotection on particle size stability and preservation of chitosan nanoparticles with and without hyaluronate or alginate coating. Saudi pharmaceutical journal : SPJ : the official publication of the Saudi Pharmaceutical Society, 25(6), 861–867. https://doi.org/10.1016/j.jsps.2016.12.008 mentions the effect of cryoprotectant and freeze drying on the stability of the chitosan nanoparticles.
3. Gutiérrez-Ruíz, S. C., Cortes, H., González-Torres, M., Almarhoon, Z. M., Gürer, E. S., Sharifi-Rad, J., & Leyva-Gómez, G. (2024). Optimize the parameters for the synthesis by the ionic gelation technique, purification, and freeze-drying of chitosan-sodium tripolyphosphate nanoparticles for biomedical purposes. Journal of biological engineering, 18(1), 12. https://doi.org/10.1186/s13036-024-00403-w mentions the synthesis of chitosan nanoparticles and the importance of freeze-drying or lyophilization on the size and stability of synthesized nanoparticles.
4. Farhank Saber Braim, Nik Noor Ashikin Nik Ab Razak, Azlan Abdul Aziz, Layla Qasim Ismael, Bashiru Kayode Sodipo, Ultrasound assisted chitosan coated iron oxide nanoparticles: Influence of ultrasonic irradiation on the crystallinity, stability, toxicity and magnetization of the functionalized nanoparticles, Ultrasonics Sonochemistry, Volume 88, 2022, 106072, ISSN 1350-4177, https://doi.org/10.1016/j.ultsonch.2022.106072 mentions the effect of ultrasonication on the stability of chitosan coated iron oxide nanoparticles.
5. Czechowska-Biskup, R., Rokita, B., Ulański, P., & Rosiak, J. M. (2015). Preparation of gold nanoparticles stabilized by chitosan using irradiation and sonication methods. Progress on Chemistry and Application of Chitin and its Derivatives, (20), 18-33. This article mentions the use of sonication to synthesize and stabilize the chitosan - gold nanoparticles.

---

## Round 0.2 · accepted · Accept

All the reviewer comments have been addressed to satisfaction.

·

Basic reporting

I have reviewed the resubmitted manuscript and commend the authors for addressing the feedback from the initial review. The revisions have significantly enhanced the clarity and overall quality of the manuscript. The revised manuscript is well-prepared and presents no major issues requiring further revisions.

Experimental design

No comment

Validity of the findings

No comment